# Risk factors for microbiologic failure in children with *Enterobacter* species bacteremia

Juri Boguniewicz[1¤]*, Paula A. Revell[2,3], Michael E. Scheurer[4], Kristina G. Hulten[1], Debra L. Palazzi[1]

1 Department of Pediatrics, Section of Infectious Diseases, Baylor College of Medicine, Houston, Texas, United States of America, 2 Department of Pediatrics, Baylor College of Medicine, Houston, Texas, United States of America, 3 Department of Pathology, Baylor College of Medicine, Houston, Texas, United States of America, 4 Department of Pediatrics, Section of Hematology/Oncology, Baylor College of Medicine, Houston, Texas, United States of America

¤ Current address: Division of Infectious Diseases, Department of Pediatrics, University of Colorado School of Medicine, Aurora, Colorado, United States of America

* juri.boguniewicz@childrenscolorado.org

## Abstract

### Background

*Enterobacter* species are an important cause of healthcare-associated bloodstream infections (BSI) in children. Up to 19% of adult patients with *Enterobacter* BSI have recurrence of infection resistant to third-generation cephalosporins (3GCs) while on therapy with a 3GC. Data are lacking regarding the incidence of and risk factors for recurrence of infection in children with *Enterobacter* BSI.

### Methods

We conducted a retrospective case-control study of patients aged ≤21 years old admitted to Texas Children's Hospital from January 2012 through December 2018 with *Enterobacter* BSI. The primary outcome was microbiologic failure from 72 hours to 30 days after the initial BSI (cases). The secondary outcome was isolation of a 3GC non-susceptible *Enterobacter sp.* from a patient with an initial 3GC-susceptible isolate.

### Results

Twelve patients (6.7%) had microbiologic failure compared to 167 controls without microbiologic failure. Of the 138 patients (77.1%) with an *Enterobacter sp.* isolate that was initially susceptible to 3GCs, 3 (2.2%) developed a subsequent infection with a non-susceptible isolate. Predictors of microbiologic failure were having an alternative primary site of infection besides bacteremia without a focus or an urinary tract infection (OR, 9.64; 95% CI, 1.77–52.31; *P* < 0.01) and inadequate source control (OR, 22.16; 95% CI, 5.26–93.36; *P* < 0.001).

**Data Availability Statement:** All de-identified data files are available from the Harvard Dataverse database (DOI: 10.7910/DVN/J75J8G).

**Funding:** The authors received no specific funding for this work.

**Competing interests:** The authors have declared that no competing interests exist.

## Conclusions

Source of infection and adequacy of source control are important considerations in preventing microbiologic failure. *In-vitro* susceptibilities can be used to select an antibiotic regimen for the treatment of *Enterobacte*r BSI in children.

## Introduction

*Enterobacter* species are an important cause of healthcare-associated bloodstream infections (BSI) in children. In pediatric-specific studies, *Enterobacter sp.* were the sixth most common cause of nosocomial BSI [1,2]. Prior studies identified young age, chronic medical comorbidities and critical illness as risk factors for *Enterobacter* BSI in children, underlining the importance of this pathogen as a cause of healthcare-associated infections [1,3–5]. Prevalence estimates based on single-center studies suggest *Enterobacter* BSI in children are rare, occurring in 0.27 to 0.44 per 1000 admissions, but associated with a high case-fatality rate, ranging from 10% to 18% [1,3–5].

Treatment of *Enterobacter* infections is complicated by the presence of an inducible AmpC beta-lactamase enzyme that is capable of inactivating certain beta-lactam antibiotics, including third-generation cephalosporins (3GCs) [6]. Antibiotic therapy can select for mutant variants that constitutively produce high levels of AmpC beta-lactamase. This selection can lead to development of resistance to most beta-lactam antibiotics, with the exception of carbapenems and cefepime, and ultimately treatment failure [6,7]. This phenomenon has typically been described with treatment using 3GCs. Studies in adults reported the development of resistance in 1 to 19% of patients on therapy with 3GC for BSI and that the emergence of resistance was associated with increased mortality, length of hospitalization and hospital charges [8–10].

Previous studies evaluating recurrence of infection in children with *Enterobacter* BSI have been limited to small, single-center studies or to discrete high-risk populations. Boyle and colleagues evaluated 45 children with *Enterobacter* BSI at a single center and found that 2 (4.4%) patients developed a resistant isolate while on therapy with a 3GC, though one patient was treated with a 3GC for a 3GC-resistant *Enterobacter* [11].

Given the paucity of data in children, we sought to determine how often recurrence of infection and development of resistance on therapy occurs with *Enterobacter* BSI and to identify risk factors for this phenomenon.

## Material and methods

### Setting and study population

The study was conducted at Texas Children's Hospital (TCH), a free-standing, tertiary care, pediatric hospital system located in the greater Houston area with a total of 815 beds. Subjects were identified from a TCH clinical microbiology laboratory database containing all patients with *Enterobacter sp.* infections from 2012–2018. Patients aged ≤18 years old admitted between January 1, 2012 and December 31, 2018 with *Enterobacter sp.* isolated from a blood culture were included. Patients with *Enterobacter* BSI who were lost to follow up within 30 days, those >18 years of age, those not treated for their bacteremia and those only with postmortem blood cultures were excluded. This study was approved by Baylor College of Medicine's Institutional Review Board (approval number H-42231). A waiver of consent was

granted by the Institutional Review Board given the retrospective nature of the study and minimal risk to participants.

## Species identification and susceptibility testing

All microbiologic specimens were processed by the TCH clinical microbiology laboratory. Species level identification was made using the Vitek 2 system, VITEK® MS (BioMérieux, Marcy-l'Étoile, France), or the Accelerate Pheno system (Accelerate Diagnostics, Tuscon, Arizona). Antibiotic susceptibility testing (AST) was performed using the E-test method, VITEK® 2 GN81 AST cards, or Accelerate Pheno system. AST results were interpreted according to the Clinical Laboratory Standards Institute M100-S(20) edition (CLSI) criteria [12]. Susceptibility to all 3GCs was inferred from reported susceptibility to cefotaxime. Isolates with intermediate or resistant susceptibility to a given antibiotic were considered non-susceptible. Molecular detection of AmpC beta-lactamases or extended-spectrum beta-lactamases (ESBLs) was not routinely done by the clinical microbiology laboratory as it is not routinely recommended by CLSI guidelines.

## Study design and data collection

A retrospective, hospital-based, case-control study was performed to determine risk factors for recurrence of infection. The primary outcome was microbiologic failure, defined as a second positive culture from any site (e.g. blood, body fluid, etc.) growing the same *Enterobacter sp.* from 72 hours to 30 days after the initial positive blood culture. Cases were patients with *Enterobacter* BSI who developed microbiologic failure. Controls were patients with *Enterobacter* BSI who did not develop microbiologic failure. The secondary outcome was isolation of a 3GC non-susceptible *Enterobacter sp.* from a patient with an initial 3GC-susceptible isolate.

Data on patient demographics, clinical variables and treatments received were abstracted from patient medical records using the REDCap electronic data capture tool [13]. Clinical variables collected included underlying comorbidities, and their severity were categorized based on the McCabe-Jackson classification [14]. Other variables obtained included antibiotic use in the prior 30 days, hospital unit, admission to the intensive care unit within 48 hours of drawing the initial positive blood culture, primary site of infection, and presence of a central venous catheter. For patients admitted to the intensive care unit, the Pediatric Index of Mortality 3 (PIM-3) score was calculated [15]. Severity of illness at the time of BSI was calculated for all patients using the Pitt bacteremia score (PBS) [16]. This study was approved by Baylor College of Medicine's Institutional Review Board.

## Definitions

Primary site of infection (e.g. bacteremia, urinary tract infection) was based on established criteria from the Centers for Disease Control and Prevention (CDC) and the National Healthcare Safety Network (NHSN) [17]. Empiric therapy was defined as antibiotic therapy received prior to the availability of susceptibility results. Empiric therapy was considered appropriate if the patient received at least one antibiotic with activity against the *Enterobacter sp.* based on susceptibility results [14]. Definitive therapy was defined as antibiotic therapy received after the availability of susceptibility results. Definitive therapy was considered appropriate if 1) the patient's isolate was susceptible to the chosen antibiotic based on in-vitro susceptibility testing, 2) the antibiotic was administered intravenously and 3) the dosage was appropriate based on the patient's weight and age. Patients who received a second antibiotic either concurrently or were changed to an alternative antibiotic during the definitive therapy course were considered to have received combination therapy [18]. Primary therapy was defined as the antibiotic

given for at least 50% of the definitive treatment course. If a patient received two or more antibiotics for at least 50% of the definitive treatment course, the patient was considered to have received combination primary therapy.

Adequate source control referred to any intervention that physically controlled a foci of infection and restored optimal function [19,20]. For example, a patient who continued to have bacteremia in the setting of having an infected central line or soft tissue abscess was not considered to have adequate source control until the infected catheter was removed or the abscess was surgically drained.

## Genotyping

Aliquoted cultures were stored at -70˚C in glycerol. For case patients with microbiologic failure and available, stored, subsequent isolates of the same *Enterobacter* species, initial and subsequent isolates were compared with pulsed-field gel electrophoresis (PFGE) using standard methods, with a few modifications and previously published electrophoresis conditions [21]. DNA was digested using XbaI and electrophoresis was performed using a CHEF DRII (Bio-Rad). In some cases, isolates were no longer available for genotyping.

## Statistical analysis

Patient demographics, clinical, microbiologic and treatment data were tabulated for all patients. Categorical variables were evaluated using Chi-square or Fisher's exact test. Continuous variables were presented with mean and standard deviation or median and interquartile range depending on the distribution of the data. Continuous variables were evaluated using Wilcoxon rank-sum test or *t*-test.

Bivariate logistic regression was performed to determine independent risk factors for microbiologic failure. Odds ratios with a 95% confidence interval and *p*-values were calculated for all possible predictors of relapse of infection. For bivariable analysis of definitive antibiotic therapy, carbapenem and cefepime were combined as the reference category (i.e. standard of care).

Variables with a *p*-value of <0.2 on bivariable analysis were assessed in an exploratory multivariable model using stepwise multivariable logistic regression starting with variables with the strongest measure of association with relapse. Goodness-of-fit was assessed with the likelihood ratio test using the models created in the stepwise process. A two-tailed *p*-value of <0.05 was considered statistically significant in all analyses. Data analysis was performed using STATA 15.1 (StataCorp LLC, College Station, Texas).

## Results and discussion

### Study population

During the 7-year study period, 212 unique patients with *Enterobacter* BSI were identified from the TCH clinical microbiology laboratory database (Fig 1). Twelve patients (6.7%) experienced microbiologic failure and were compared to 167 control patients without microbiologic failure. Of those who developed microbiologic failure, 8 patients (4.5%) had relapse of bacteremia, 2 of which also had infection at a secondary site (biliary, 1; intraabdominal, 1). Four patients (2.2%) had relapse of infection exclusively at secondary sites: urine, 1; peritoneal fluid, 1; intrapelvic abscess, 1; and tracheal aspirate, 1. The median time to microbiologic failure was 4.5 (range 3–22) days after the initial positive blood culture. Three case patients had persistence of their bacteremia and 9 had relapse of infection after clearing their bacteremia. Of the 138 patients (77.1%) with an *Enterobacter sp*. isolate that was initially susceptible to

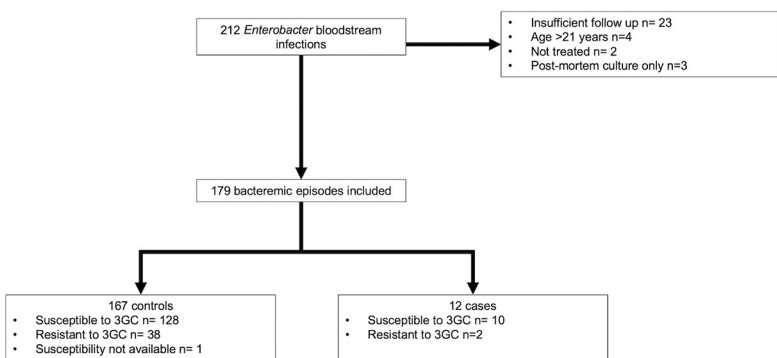

**Fig 1. Study selection flowchart.** 3GC, third generation cephalosporin.

3GCs, 3 (2.2%) developed a subsequent infection with a non-susceptible isolate (2 relapsed bacteremia, 1 relapsed tracheostomy infection). One case patient died.

## Risk factors for microbiologic failure after *Enterobacter* species bacteremia

Among the entire study cohort, the median age was 16.7 months (interquartile range, 5.8 to 63 months); 52% were male. All cases and 93% of controls had at least one underlying medical condition (Table 1). On bivariable analysis, underlying hepatobiliary disease (OR, 8.94; 95% CI, 1.21–49.88; $P = 0.001$) was associated with microbiologic failure. The most common primary site of infection was bacteremia (82.1%) followed by urinary tract infection (UTI) (9.5%) and intra-abdominal infection (3.4%). Compared to controls, cases had a higher odds of having an alternative site of infection besides bacteremia or a UTI (OR, 7.09; 95% CI, 1.32–31.45; $P = 0.011$). Cases were evenly distributed over the 7-year study period. Further details of cases with microbiologic failure are described in S1 Table.

The most common species identified was *Enterobacter cloacae* (83.8%) followed by *Klebsiella aerogenes* (formerly *Enterobacter aerogenes*) (12.3%); and other *Enterobacter* species (2.2%); in 3 patients (1.7%) the species was not determined. Forty-one patients (22.9%) had an initial *Enterobacter sp*. bloodstream isolate that was non-susceptible to 3GCs. Another 47 (26.3%) had a polymicrobial infection. There were no statistically significant differences between cases and controls in regards to the bacterial species, having a 3GC non-susceptible isolate or having a polymicrobial infection. The median duration of bacteremia was 1 day (IQR 1–2) in both groups.

The majority of patients (159/179, 88.9%) received appropriate empiric antibiotic therapy, most frequently with a combination of a beta-lactam and an aminoglycoside (48.6%) followed by piperacillin-tazobactam (18.4%), 3GC (14.5%) and cefepime (11.7%) monotherapy (Table 2). A 3GC was most commonly used for definitive (30.7%) and primary therapy (36.9%). The definitive antibiotic regimen was appropriate in nearly all patients (97.2%). However, a narrower-spectrum agent could have been selected based on *in-vitro* susceptibilities in 50.2%. Cases and controls did not differ in median duration of definitive treatment or definitive or primary antibiotic regimen, particularly regarding the use of a 3GC. However, cases were more likely to have inadequate source control (OR, 18.88; 95% CI, 4.34–92.43; $P < 0.001$). Four of the 8 case patients without adequate source control had complicated intraabdominal infections or wound infections and did not undergo operative procedures to establish source control until after relapse. The majority of patients (83%) had a central line and median time to removal was longer in cases than controls ($P = 0.06$). None of the cases compared to 78% of controls had removal of the central line within 72 hours of onset of bacteremia ($P < 0.001$).

**Table 1. Demographic and clinical risk factors for microbiologic failure in children with *Enterobacter* bacteremia and bivariable analysis.**

| Characteristic | Cases[a] (n = 12) | Controls (n = 167) | P-value |
|---|---|---|---|
| Demographics | | | |
| Age, median (IQR), years | 1.4 (0.6–3.7) | 1.5 (0.5–6) | 0.503 |
| Male sex | 4 (33) | 97 (60) | 0.132 |
| Hispanic ethnicity | 8 (67) | 62 (37) | 0.043 |
| Black race | 3 (25) | 38 (23) | 0.989 |
| Comorbidities | | | |
| Hematologic malignancy | 1 (8) | 31 (19) | 0.372 |
| Bone marrow transplant | 0 | 20 (12) | 0.203 |
| Solid malignancy | 1 (8) | 12 (7) | 0.882 |
| Short gut syndrome | 2 (17) | 27 (16) | 0.964 |
| Prematurity | 4 (33) | 36 (22) | 0.344 |
| Gestational age, median (IQR), weeks | 25 (24–31) | 27 (25–33) | 0.354 |
| Congenital heart disease | 1 (8) | 22 (13) | 0.628 |
| Chronic lung disease | 1 (8) | 14 (8) | 0.995 |
| Hepatobiliary disease | 3 (25) | 6 (4) | <0.001 |
| Solid organ transplant | 1 (8) | 7 (4) | 0.502 |
| Immunosuppression | 4 (33) | 64 (38) | 0.731 |
| Neutropenia | 0 | 27 (16) | 0.131 |
| Prior hospitalization in past 6 months | 9 (75) | 110 (66) | 0.518 |
| LOS before bacteremia, median (IQR), days | 5 (1–15) | 3 (0–26) | 0.941 |
| Hospital acquired infection | 8 (67) | 83 (50) | 0.256 |
| ICU stay prior to bacteremia | 5 (42) | 72 (43) | 0.922 |
| LOS in ICU, median (IQR), days | 36 (7–138) | 29 (3–108) | 0.703 |
| PIM-3 score, median (IQR) | 3.17 (1.60–70.51) | 4.45 (1.69–19.51) | 0.780 |
| Mechanical ventilation | 4 (33) | 32 (19) | 0.237 |
| Pitt Bacteremia Score, mean (SD) | 1.5 (0–4) | 1 (0–2) | 0.887 |
| Central line present | 10 (83) | 139 (83) | 0.993 |
| Foley catheter present | 1 (8) | 6 (4) | 0.413 |
| Primary site of infection[b] | | | |
| Bacteremia | 8 (67) | 139 (83) | 1 (ref) |
| Urinary tract | 0 | 17 (10) | 1 (ref) |
| Intra-abdominal[c] | 3 (25) | 5 (3) | <0.001 |
| Surgical site infection | 1 (8) | 1 (1) | 0.005 |
| *Enterobacter cloacae* infection[d] | 10 (83) | 140 (84) | 0.964 |
| Baseline isolate 3GC non-susceptible | 2 (17) | 39 (23) | 0.594 |
| Polymicrobial infection | 3 (25) | 44 (26) | 0.918 |

OR, odds ratio; IQR, interquartile range; PIM-3, Pediatric Index of Mortality 3; SD, standard deviation.

[a]Unless otherwise indicated, data represent number (percentage).

[b]Bacteremia and UTI combined as the reference group.

[c]Intra-abdominal infections included any deep-seated intra-abdominal infection (e.g., abscess) as well as hepatobiliary infections (e.g., cholangitis).

[d]There were missing values for species (n = 3).

## Multivariable analysis

After adjusting for definitive therapy, sex, ethnicity and underlying hepatobiliary disease, the strongest predictors of microbiologic failure were having an alternative primary site of infection besides bacteremia or a UTI (OR, 9.64; 95% CI, 1.77–52.31; $P < 0.01$) and inadequate source control (OR, 22.16; 95% CI, 5.26–93.36; $P < 0.001$).

**Table 2. Treatment regimens and outcomes.**

| Treatment variable | Cases (n = 12)[a] | Controls (n = 167) | P-value |
|---|---|---|---|
| Empiric antibiotic | | | |
| Carbapenem/Cefepime[b] | 1 (8) | 23 (14) | 1 (ref) |
| Aminoglycoside | 1 (8) | 7 (4) | 0.399 |
| Fluoroquinolone | 1 (8) | 0 | <0.001 |
| Piperacillin-tazobactam | 4 (33) | 29 (17) | 0.295 |
| Third generation cephalosporin[c] | 2 (17) | 24 (14) | 0.600 |
| Combination | 3 (25) | 84 (50) | 0.867 |
| Empiric therapy inappropriate | 1 (8) | 19 (11) | 0.895 |
| Definitive antibiotic | | | |
| Carbapenem/Cefepime[b] | 5 (42) | 63 (37) | 1 (ref) |
| Aminoglycoside | 0 | 8 (5) | 0.428 |
| Fluoroquinolone | 0 | 7 (4) | 0.458 |
| Piperacillin-tazobactam | 2 (17) | 16 (10) | 0.604 |
| Third generation cephalosporin | 3 (25) | 52 (31) | 0.671 |
| Combination | 2 (17) | 21 (13) | 0.835 |
| Duration of definitive therapy, median (IQR), days | 9 (4–14) | 9 (7–12) | 0.570 |
| Definitive therapy appropriate | 12 (100) | 162 (97) | 0.543 |
| Definitive therapy most narrow spectrum option | 5 (42) | 84 (50) | 0.564 |
| Primary antibiotic | | | |
| Carbapenem/Cefepime | 2 (17) | 52 (31) | 1 (ref) |
| Aminoglycoside | 0 | 10 (6) | 0.536 |
| Fluoroquinolone | 1 (8) | 10 (6) | 0.438 |
| Piperacillin-tazobactam | 2 (17) | 19 (11) | 0.314 |
| Third generation cephalosporin[c] | 6 (50) | 60 (36) | 0.239 |
| Combination | 1 (8) | 16 (10) | 0.697 |
| Central line not removed[d] | 10 (100) | 109 (78) | 0.100 |
| Source control inadequate | 8 (67) | 16 (10) | <0.001 |

IQR, interquartile range.

[a]Unless otherwise indicated, data represent number (percentage).

[b]Patients receiving carbapenems or cefepime were combined as the reference group.

[c]Third generation cephalosporins included ceftriaxone, cefotaxime and ceftazidime.

[d]Population at risk (i.e. central line present at time of bacteremia) for cases (n = 10) and controls (n = 139).

## Genotyping

Isolates were available for genotyping by PFGE for 7 of 12 cases with microbiologic failure. PFGE revealed identical or closely related banding patterns between initial and subsequent isolates in 6 patients (86%) indicating relapse rather than newly acquired infection. However, interstrain comparisons between patients revealed genetic diversity, and no dominant PFGE profile was observed across patient isolates.

## Discussion

Our findings suggest that in children with *Enterobacter* BSI, microbiologic failure and development of resistance to 3GCs while on therapy is rare. Source of infection was identified as an important risk factor for microbiologic failure, and patients with a primary site of infection besides bacteremia or a urinary tract infection had higher odds of relapsed infection.

Furthermore, our results emphasize the importance of source control in preventing microbiologic failure in patients with *Enterobacter* BSI.

To the best of our knowledge, this study represents the largest evaluating *Enterobacter* BSI in children and highlights unique risk factors for microbiologic failure in pediatric patients. Previous studies describing outcomes of children with *Enterobacter* BSI have been limited to case series or focused on discrete high-risk patient groups such as premature neonate and have not focused on specific risk factors for microbiologic failure [3,11,22–24]. In addition, our study considered source control as a risk factor for microbiologic failure and genotyped available isolates to confirm relapse of infection.

Similar to other pediatric studies that identified young age and chronic medical conditions as risk factors for *Enterobacter* BSI, the vast majority of our cohort consisted of younger children and nearly all patients had at least one chronic medical condition [1,3–5]. Our finding that 6.7% of pediatric patients with *Enterobacter* BSI developed microbiologic failure is similar to results from a recent study of 26 critically-ill children by Alharjri et al. [25]. Our estimate that 2.2% of patients with an initially 3GC-susceptible isolate developed resistance to 3GCs while on therapy is identical to the findings of Boyle et al. [11] but much lower than some seminal studies in adults [8,10,26]. A recent study of 922 adults with *Enterobacter* BSI found that relapse of infection occurred in 3.4% of patients and resistance to 3GC emerged during treatment in 3 (<1%) patients [18].

Previous studies in adults have identified treatment with 3GCs as an important risk factor for the development of resistance while on therapy and relapse of infection [8,10,26]. In contrast, treatment with 3GCs was common at our institution and was not associated with microbiologic failure in our study. There are a number of possible explanations for this discrepancy. First, underlying, chronic comorbidities were different in our pediatric study population compared to those in adults. In Choi et al.'s study, 6 of 14 patients who developed relapse of infection had biliary tract infections related to an unresectable bile duct malignancy [26]. Such malignancies are rare in children [27]. In addition, the authors found that such infections were often incompletely drained suggesting inadequate source control was a potential confounding factor. Kaye et al. found that 19% of adult patients developed resistance to 3GCs while on therapy, though genotyping of subsequent isolates was not performed and, in some cases, subsequent isolates represented a different species [8]. It is also possible that relapses in previous studies could have been due to a common clone. Though isolates were available for only 7 of our case patients, there was significant interpatient genetic diversity suggesting a common clone was not being transmitted between patients. Previous studies citing higher rates of relapse of infection were also conducted before CLSI breakpoints for ceftriaxone for *Enterobacter sp*. were lowered in 2010. Consequently, some patients in these studies may have been treated with suboptimal therapy. Our data suggest that the revised CLSI breakpoints are appropriate for predicting clinical response to 3GCs in patients with *Enterobacter* BSI.

Our finding that inadequate source control is a major risk factor for microbiologic failure is consistent with a recent study by Harris et al. in which nearly 75% of relapsed cases had central-line associated bacteremia (CLABSI) where a delay in catheter removal was hypothesized as the cause of relapse [18]. Similarly, Tamma et al. suggested that cefepime would be an adequate choice in the treatment of AmpC-producing *Enterobacteriaciae* when adequate source control is possible [20]. However, the vast majority of patients (>93%) in this study had adequate source control so the investigators were not able to draw firm conclusions about patients who did not have adequate source control. In our study, 83% of patients had a central line and median time to removal was longer in cases than controls. We found that bacteremia was most common in our cohort. In the majority of these patients, bacteremia was likely related to an infected central line or intestinal translocation. It is possible that children with bacteremia due

to intestinal translocation had low bacterial inoculum in the bloodstream which may have contributed to the success of therapy regardless of antibiotic choice.

The Infectious Diseases Society of America CLABSI guideline and several expert opinions recommend routine use of carbapenems for the treatment of *Enterobacter* BSI with cefepime as the preferred alternative [28–30]. However, our data suggest that in children with uncomplicated bacteremia or with a urinary source due to *Enterobacter* sp., the use of 3GCs for definitive therapy may not be associated with relapse of infection if source control can be achieved. This practice could help reduce carbapenem and cefepime overuse.

Our study has a number of limitations related to its retrospective and observational design. This was a single-center study and our rates of 3GC-resistance may not be reflective of other institutions. Though the power of our study was limited by the sample size, to the best of our knowledge, this is the largest pediatric study to evaluate risk factors for microbiologic failure in *Enterobacter sp*. BSI. The heterogeneity of antibiotic regimens makes it difficult to determine if any single antibiotic was associated with microbiologic failure. However, nearly a third of patients were treated with 3GCs and none with adequate source control had relapse of infection, suggesting that 3GCs are a reasonable treatment option for 3GC-susceptible *Enterobacter sp*. infections when source control is possible. It is also possible that case patients may have been misclassified if their subsequent isolate represented a newly acquired infection rather than relapse of the same clone. This seems unlikely given the genotyping results of available isolates. In addition, one case patient who developed resistance to 3GCs while on therapy only had relapse at a tracheal aspirate site. While the clinical significance of this is debatable, he met criteria for a possible ventilator associated pneumonia and was treated as such. Regardless, not including this case would only have strengthened our conclusion that microbiologic failure and emergence of resistance while on therapy is rare in children. In addition, we did not perform molecular analysis to confirm the mechanism of resistance in relapsed cases that developed resistance to 3GCs while on therapy. While other mechanisms of resistance, such as co-existing ESBLs could have accounted for such resistance rather than an AmpC beta-lactamase, all *Enterobacter* species chromosomally encode for an AmpC beta-lactamase and have the potential for inducible resistance via this mechanism when exposed to an appropriate antibiotic substrate. Of greater concern clinically is the selection of stably de-repressed mutants that constitutively express the AmpC beta-lactamase leading to treatment failure. Regardless of the mechanism of resistance, our data suggest that this is a rare phenomenon in children, occurring in 3 patients of 138 with an isolate that was initially susceptible to 3GCs. Finally, most patients in our cohort with central lines had their lines removed promptly which could have contributed to lower relapse rates. We were not able to evaluate whether patients received other infection preventative measures after diagnosis of a central-line associated BSI that may have impacted the risk of recurrence. However, these institutional practices would not have been expected to differ between cases and controls.

A large, multi-center, prospective study of *Enterobacter* BSI is needed to determine the optimal treatment regimen for these infections in children. Other sites of infection not associated with bacteremia should also be evaluated as risk factors for relapse may differ, though source control is likely an important aspect of treatment. Finally, though our findings suggest central line removal may be important in preventing microbiologic failure in patients with *Enterobacter* BSI, data about the efficacy of antibiotic lock therapy or other infection preventative measures are lacking.

## Conclusions

In summary, in a cohort of children with *Enterobacter sp*. bacteremia, microbiologic failure and development of resistance to 3GCs while on therapy was rare. Source of infection and

adequacy of source control are important considerations in preventing microbiologic failure. Based on our results, *in-vitro* susceptibilities using current CLSI breakpoints seem reliable and can be used to select tailored antibiotic regimens for the treatment of such infections.

## Supporting information

**S1 Table. Demographic and clinical characteristics of pediatric patients with microbiologic failure following *Enterobacter spp.* bacteremia.** CLABSI (Central-line associated bloodstrea infection); CVC (central venous catheter); mo (months); VAP (ventilator associated pneumonia); SSI (surgical site infection); RV (right ventricle); PA (pulmonary artery).
(DOCX)

## Acknowledgments

We thank Drs. Sheldon L. Kaplan, Timothy Palzkill, Jesus G. Vallejo, Judith Campbell, Tao Wang and Lynn Zechiedrich for their advice on the study design and analysis.

## Author Contributions

**Conceptualization:** Juri Boguniewicz, Paula A. Revell, Debra L. Palazzi.

**Data curation:** Juri Boguniewicz.

**Formal analysis:** Juri Boguniewicz, Kristina G. Hulten.

**Methodology:** Juri Boguniewicz, Paula A. Revell, Michael E. Scheurer, Kristina G. Hulten, Debra L. Palazzi.

**Resources:** Kristina G. Hulten.

**Supervision:** Kristina G. Hulten, Debra L. Palazzi.

**Writing – original draft:** Juri Boguniewicz.

**Writing – review & editing:** Juri Boguniewicz, Paula A. Revell, Michael E. Scheurer, Kristina G. Hulten, Debra L. Palazzi.

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
