## [Decision Letter · Decision Letter 0]

5 Jul 2021

PONE-D-21-14421

Risk factors for microbiologic failure in children with Enterobacter species bacteremia

PLOS ONE

Dear Dr. Boguniewicz,

Thank you for submitting your manuscript to PLOS ONE. After careful consideration, we feel that it has merit but does not fully meet PLOS ONE’s publication criteria as it currently stands. Therefore, we invite you to submit a revised version of the manuscript that addresses the points raised during the review process.

The reviewer has raised number of questions that need clarifications and explanations. Please address all points raised by the reviewer.

We look forward to receiving your revised manuscript.

Kind regards,

Iddya Karunasagar

Academic Editor

PLOS ONE

Journal Requirements:

Additional Editor Comments (if provided):

The reviewer has pointed out a number of aspects of the manuscript that need clarifications, justifications and further explanation. Please address all comments point by point.

Reviewers' comments:

Reviewer's Responses to Questions

**Comments to the Author**

1. Is the manuscript technically sound, and do the data support the conclusions?

Reviewer #1: Yes

2. Has the statistical analysis been performed appropriately and rigorously? 

Reviewer #1: Yes

3. Have the authors made all data underlying the findings in their manuscript fully available?

Reviewer #1: Yes

4. Is the manuscript presented in an intelligible fashion and written in standard English?

Reviewer #1: Yes

5. Review Comments to the Author

Reviewer #1: Dear Authors

The manuscript describes Enterobacter infections in a tertiary paediatric set up including the risk factors for microbiological failure and development of subsequent resistance. The manuscript has been well presented and highlights an important topic both in Infection control and infectious diseases and clinical microbiology.

However, a few clarifications can better the manuscript greatly

1. How was ESBL and AmpC detection done in the study? The authors explain 3rd gen cephalosporins but do not explain which 3rd gen cephalosporin was used for testing or treatment. What was the general percentage of AmpC in the hospital? Did that change over the study period?

2. The methodology defines including all cases less than 21 years who developed infection with Enterobacter species. The supplementary table shows most cases (except 3 cases) in children less than 4 years. What was the possible source for the organism. Was there a possibility of an outbreak with more than 2 cases occurring around the same time? Or was there a constant reservoir in the hospital?

3. Since most were associated with CLABSI, what were the infection control measures put in place to reduce the occurrance?

4. What was the antibiogram of the hospital and what was the basis of the choice of empirical antibiotics described in the study? What was the data on escalation and de-escalation of antibiotics and did they impact outcomes?

5. Table 1 on the demographis and risk factors classifies hepatobiliary disease and intraabdominal/hepatobiliary seperately. Is there a reason for the same?

6. Materials and methods describes genotyping as if done on all isolates. But only 7 isolates were chosen. What was the basis of the choice of isolates?

6. PLOS authors have the option to publish the peer review history of their article (what does this mean?). If published, this will include your full peer review and any attached files.

Reviewer #1: No

---

## [Author Response · Author response to Decision Letter 0]

19 Aug 2021

Iddya Karunasagar, PhD

Academic Editor

PLOS ONE

August 19, 2021

Dear Dr. Karunasagar,

We thank you for inviting us to submit a revised version of our manuscript entitled, “Risk factors for microbiologic failure in children with Enterobacter species bacteremia” for publication in PLOS ONE. We are grateful to you and the reviewer for taking the time to provide us with useful and constructive feedback on our manuscript. We have taken all of the recommendations provided into consideration and have made substantial changes to our manuscript, which we believe significantly strengthen our paper. We hope that these revisions and the responses provided address all the issues and concerns you and the reviewer have raised. Below is a point-by-point response to the questions and comments enumerated in your decision letter. 

Editor comments:

1. Please ensure that your manuscript meets PLOS ONE’s style requirements, including those for file naming.

Thank you for your comment and including the relevant links for PLOS ONE’s formatting templates. These templates and the formatting requirements have been reviewed in detail to ensure our manuscript meets the style requirements. The supplemental table file name has been updated in accordance with these requirements. File names for the components of the resubmission have been named according to the recommendations outlined on the PLOS ONE website for Revising Your Manuscript. 

2. Please include captions for your Supporting Information files at the end of your manuscript, and update any in-text citations to match accordingly. 

Thank you for pointing out this omission. A caption for Supplemental Table 1 has also been added at the end of the manuscript body under the level 1 heading “Supporting information,” lines 477-480. In-text citations for this supplemental information were verified to comply with the Journal’s formatting requirements. 

3. Thank you for including your ethics statement on the online submission form: "This study was approved by Baylor College of Medicine’s Institutional Review Board. Approval number H-42231. A waiver of consent was granted by the Institutional Review Board given the retrospective nature of the study and minimal risk to participants.". To help ensure that the wording of your manuscript is suitable for publication, would you please also add this statement at the beginning of the Methods section of your manuscript file.

Thank you for bringing this to our attention. The wording above has been added to the first paragraph of the Methods section, lines 97-100.

Reviewer 1 comments:

1. How was ESBL and AmpC detection done in the study? The authors explain 3rd gen cephalosporins but do not explain which 3rdgen cephalosporin was used for testing or treatment. What was the general percentage of AmpC in the hospital? Did that change over the study period? 

We appreciate Reviewer 1’s comments. We agree verifying the mechanism of resistance would be interesting. However, genotyping of Amp-Cs is not routinely performed for clinical 

specimens at our institution in accordance with CLSI guidelines. Additionally, as Enterobacter species chromosomally encode an inducible AmpC beta-lactamase, the potential for AmpC-mediated resistance was assumed for all Enterobacter species. A sentence clarifying this has been added to the methods section in lines 110-112. Similarly, we are not able to estimate the percentage of AmpC producing organisms or ESBLs in our hospital or how this may have changed over time as testing is not routinely performed by the clinical microbiology laboratory. An acknowledgment and explanation of this limitation has also been added to the discussion section, lines 356-365. 

We agree the term “third-generation cephalosporins” could cause confusion as different institutions use various agents of this antibiotic class and some agents are no longer available (i.e, cefotaxime). Antibiotic susceptibility testing for all 3rd generation cephalosporins was inferred from the results of ceftriaxone testing and a sentence clarifying this has been added to the methods section, line 108. A footnote “c” has also been added to Table 1 to indicate that third-generation cephalosporins refers specifically to ceftriaxone, cefotaxime or ceftazidime, line 259. Column headings in Supplemental Table 1 have also been updated to specify whether the initial isolate was susceptible to ceftriaxone rather than “3GCs” and, similarly, whether there was development of resistance to ceftriaxone rather than “3GCs.” 

2. The methodology defines including all cases less than 21 years who developed infection with Enterobacter species. The supplementary table shows most cases (except 3 cases) in children less than 4 years. What was the possible source for the organism. Was there a possibility of an outbreak with more than 2 cases occurring around the same time? Or was there a constant reservoir in the hospital?

We agree with Reviewer 1 that these are interesting questions. Prior pediatric studies have noted that young age, chronic medical comorbidities and critical illness are risk factors for Enterobacter bloodstream infections. A sentence regarding this observation has been added to the introduction, lines 64-66. Similarly, the majority of our cohort included younger patients <5 years with chronic medical comorbidities and a comment highlighting this has been added to the discussion, lines 292-294. The source of the organism is unclear though the majority of patients had healthcare-associated infections. Only 2 of the cases with microbiologic failure occurred around the same time, though this was in different hospital units making an outbreak unlikely. Cases were otherwise evenly distributed over the 7-year study period and a sentence clarifying this has been added to the results section, line 214. Where isolates were available for genotyping among case patients, there was considerable interpatient strain genetic variability suggesting a lack of a common source. 

3. Since most were associated with CLABSI, what were the infection control measures put in place to reduce the occurrence?

We thank Reviewer 1 for their comment. This is an interesting question but, unfortunately, we did not have the data to be able to comment on whether any infection prevention measures may have differed between cases and controls. However, CLABSI prevention has been a focus in our hospital, as it has been nationally, and numerous measures (e.g., insertion bundles, handwashing campaigns, etc.) have been instituted over time, though this would not be expected to have differed between cases and controls. A comment to this effect was added to the discussion, lines 367-370.

4. What was the antibiogram of the hospital and what was the basis of the choice of empirical antibiotics described in the study? What was the data on escalation and de-escalation of antibiotics and did they impact outcomes?

Seventy-seven percent of patients had an isolate that was considered “susceptible” to 3rd generation cephalosporins at baseline based on CLSI breakpoints. The heterogeneity of empiric antibiotic regimens, like definitive therapy, makes it difficult to determine if any individual agent was associated with worse or better outcomes as acknowledged in the discussion. We note that the vast majority of patients in both groups (92% cases vs. 89% controls) received “appropriate” empiric therapy as defined by an agent given by the correct route (i.e., parenterally), at the correct dose, and for which the isolate was ultimately identified as susceptible in retrospect based on antimicrobial susceptibility results. A column describing empiric therapy for cases has been added to Supplemental Table 1. Similarly, definitive therapy was deemed appropriate in 100% of cases and 97% of control patients. We also note that definitive antibiotic regimens were similar between the case and control groups, though in both groups narrower antimicrobial agents could have been selected in 42% and 50% of cases and controls respectively (i.e., de-escalated) based on susceptibility results alone. It is uncertain if results would have been similar had these patients received narrower spectrum therapy, though our results that relapse was rare in patients treated with third-generation cephalosporins is encouraging that these agents can be safely used instead of cefepime and carbapenems when the isolate is susceptible and source control is achievable. 

5. Table 1 on the demographics and risk factors classifies hepatobiliary disease and intraabdominal/hepatobiliary separately. Is there a reason for the same?

We thank the Reviewer for pointing out this confusing nomenclature. In Table 1 “hepatobiliary” is listed as an underlying comorbid condition and “intraabdominal/hepatobiliary” is listed as a primary site of infection. The term “hepatobiliary” has been removed from the primary site of infection to avoid confusion and a footnote has been added to indicate that “intraabdominal” as a primary site of infection refers to any deep-seated intraabdominal infection, including hepatobiliary infections such as cholangitis, lines 224-225. 

6. Materials and methods describes genotyping as if done on all isolates. But only 7 isolates were chosen. What was the basis of the choice of isolates?

Thank you for pointing out this potential inconsistency. The relevant materials and methods section for genotyping has been clarified to indicate genotyping was done only on cases with microbiologic failure of their Enterobacter infection who had both isolates available for genotyping (i.e., the index isolate and the isolate from the relapse of infection), lines 163-168. Genotyping was not done on control patients. Unfortunately, banked specimens were not available from the clinical microbiology laboratory for every case patient to be able to verify relapse of infection with the same strain versus a de novo infection. The genotyping section of the methods has also been moved to follow the study design and definitions sections that establish the definitions for cases and controls and microbiologic failure to avoid confusion. 

We thank you and the reviewer again for your time and thoughtful comments and the opportunity to strengthen our manuscript based on this feedback. We appreciate your reconsideration of our submission. 

Sincerely,

Juri Boguniewicz, MD

Corresponding Author

Assistant Professor of Pediatrics

University of Colorado School of Medicine

Juri.boguniewicz@childrenscolorado.org

---

## [Decision Letter · Decision Letter 1]

20 Sep 2021

Risk factors for microbiologic failure in children with Enterobacter species bacteremia

PONE-D-21-14421R1

Dear Dr. Boguniewicz,

We’re pleased to inform you that your manuscript has been judged scientifically suitable for publication and will be formally accepted for publication once it meets all outstanding technical requirements.

Kind regards,

Iddya Karunasagar

Academic Editor

PLOS ONE

Additional Editor Comments (optional):

All comments addressed satisfactorily

Reviewers' comments:

Reviewer's Responses to Questions

**Comments to the Author**

1. If the authors have adequately addressed your comments raised in a previous round of review and you feel that this manuscript is now acceptable for publication, you may indicate that here to bypass the “Comments to the Author” section, enter your conflict of interest statement in the “Confidential to Editor” section, and submit your "Accept" recommendation.

Reviewer #1: All comments have been addressed

2. Is the manuscript technically sound, and do the data support the conclusions?

Reviewer #1: Yes

3. Has the statistical analysis been performed appropriately and rigorously? 

Reviewer #1: Yes

4. Have the authors made all data underlying the findings in their manuscript fully available?

Reviewer #1: Yes

5. Is the manuscript presented in an intelligible fashion and written in standard English?

Reviewer #1: Yes

6. Review Comments to the Author

Reviewer #1: THE AUTHORS HAVE ADEQUATELY ANSWERED ALL QUERIES . THOSE THAT COULD NOT BE ADDED AS LIMITATIONS.

7. PLOS authors have the option to publish the peer review history of their article (what does this mean?). If published, this will include your full peer review and any attached files.

Reviewer #1: No

---

## [Editor Report · Acceptance letter]

29 Sep 2021

PONE-D-21-14421R1 

Risk factors for microbiologic failure in children with *Enterobacter* species bacteremia 

Dear Dr. Boguniewicz:

I'm pleased to inform you that your manuscript has been deemed suitable for publication in PLOS ONE. Congratulations! Your manuscript is now with our production department. 

Kind regards, 

on behalf of

Dr. Iddya Karunasagar 

Academic Editor

PLOS ONE